# Microsite Determines the Soil Nitrogen and Carbon Mineralization in Response to Nitrogen Addition in a Temperate Desert

Yingwu Chen *, Haorui Li, Huilu Sun and Yuxin Guo

College of Horticulture and Plant Protection, Henan University of Science and Technology, Luoyang 471023, China
* Correspondence: cyw@haust.edu.cn

**Abstract:** Nitrogen deposition can change the soil in N and C cycling processes. However, a general understanding of how N deposition changes C and N mineralization has not yet been reached. Soil organic C and N mineralization beneath the dominant shrubs of *Haloxylon ammodendron* and between the shrubs in response to two levels of N addition (2.5 gN m$^{-2}$ and 5 gN m$^{-2}$ per year) were investigated in the 1st, 4th, and 9th year of N addition in a temperate desert of northern China. N addition promoted soil N mineralization ($R_mN$), and the nitrification rate ($R_{NN}$) increased C mineralization in the interplant and decreased it beneath shrubs. N addition increased soil microbial biomass C ($C_{mic}$), N ($N_{mic}$), and PLFAs in the interplant, and decreased it beneath shrubs. $R_mN$ and $R_{NN}$ were related to $N_{mic}$, and $R_{CM}$ was related to $C_{mic}$ and the total PLFAs. N addition increased the fungal biomass alongside the ratio of fungal to bacterial PLFAs in the interplants while decreasing them beneath shrubs. Our results support how N addition can increase soil N mineralization and nitrification, but the effects on soil C mineralization are dependent on the amount of nitrogen addition, the soil's available carbon content, and water. Finally, the divergent responses of microbial communities to N addition between microsites suggest that the "fertile islands" effects on nutrients and microbial biomass are important when estimating feedbacks of C and N cycling to projected N deposition in the desert ecosystem.

**Keywords:** carbon cycling; desert; microbial biomass; nitrogen cycling; path analysis; phospholipid fatty acids; soil respiration





## 1. Introduction

The impact of atmospheric nitrogen (N) deposition on soil carbon (C) and nitrogen (N) cycling in temperate deserts remains a subject that requires further understanding due to the complex responses of soil microbial communities to increasing N levels [1–4]. Previous studies have shown the nonlinear responses of soil microbial communities to various N addition doses, indicating the potential variations in their response, which is dependent on the experimental region and N dose [5–11]. Furthermore, variations in the soil N background and microorganisms among studies, as well as the potential for N saturation, can influence microbial responses. Additionally, the indirect effects of increased N, such as soil acidification [9,12–14] and changes in the substrate ratios [4,15–17], can also influence soil microbial communities. Despite recognizing the close link between soil N cycling and microbial communities, uncertainties remain regarding the impact of the increased N input on microbial activities, the community structure, and subsequent feedback to soil N mineralization.

Increasing N levels can either enhance the coupling of C and N cycling by promoting primary production and soil C storage [18–20] or decoupling these cycles [21–24] due to reduced C:N ratios, which can accelerate the decomposition of plant litter and soil organic

matter [22,24]. Therefore, understanding the response of soil C mineralization is essential when comprehending soil N dynamics in the context of increased N deposition.

In dryland ecosystems, "fertile islands" formed by shrub habitations create significant differences in soil nutrients and the organic C content [25–27]. Various ecological processes, including soil respiration and N mineralization, exhibit distinct variations between soils beneath shrubs and interplant soil [28–30]. Nutrient heterogeneity in desert ecosystems plays a vital role in the response of C and N dynamics to increased N, surpassing other terrestrial ecosystems [31–33]. However, our understanding of the interactive effects between "fertile islands" and N deposition on soil microbial properties and N cycling in temperate deserts remains limited.

In this study, we investigated soil N mineralization, nitrification, and C mineralization in the 1st, 4th, and 9th year after simulating the N deposition of 2.5 and 5 gN m$^{-2}$ yr$^{-1}$ in the Gurbantungute desert located in northwestern China. We hypothesized that (1) N addition would significantly increase soil C and N mineralization and (2) there would be additive effects of "fertile islands" and N addition on soil C and N mineralization, as increased soil C and N availability, could enhance soil microbial activities.

## 2. Materials and Methods

### 2.1. Study Site Description

The study took place on the southern border of the Gurbantunggut Desert, situated at the heart of the Eurasian Continent (44°17′ N, 87°56′ E, and 475 m a.s.l.). This region experiences a continental arid temperate climate that is characterized by scorching, dry summers and cold winters. The average annual temperature is 6.6 °C, and the yearly average precipitation stands at 160 mm, with 70% to 80% occurring during the plant growing season, spanning from April to September. The soils in this area consist of desert solonetz, with a layer of aeolian sandy soil on the surface. Shrubbery and semi-shrubs, predominantly *Tamarix ramosissima*, *Haloxylon ammodendron*, and *H. persicum,* cover approximately 30% of the land. The herbaceous layer, which comprises ephemerals and annuals, accounts for 40% of the area and includes species such as *Alyssum linifolium*, *Leptaleum filifolium*, *Erodium oxyrrhynchum*, *Myosotis scorpioides*, *Eremurus inderiensis*, *Salicornia brachiate*, and *Ceratocarpus arenarius*. The presence of "fertile islands" leads to significant variations in soil characteristics between the areas and between the plants and regions beneath the *H. ammodendron* shrubs (see Table S1). Soil properties beneath the shrubs, such as soil organic carbon, the total nitrogen and phosphorus content, pH levels, and electronic conductivity (EC), exhibit notably higher values compared to those found in the interplant areas. Additionally, the soil bulk density of the 0–5 cm layer is lower beneath *H. ammodendron* shrubs compared to the interplant zones.

### 2.2. Experimental Design and Sample Collection

The experiment employed a completely randomized block design, consisting of six blocks that each contained three plots. These blocks were distributed across three interlands located between sandy dune belts, with two blocks situated in each interland. Within each block, the CK (N0, 0 N addition), N1 (2.5 gN m$^{-2}$ yr$^{-1}$ addition), and N2 (5 gN m$^{-2}$ yr$^{-1}$ addition) treatments were randomly assigned to the plots, resulting in a total of six replications for each treatment. Each plot had dimensions of 10 × 10 m, and a buffer zone of 10 m in width was established between the adjacent plots. Additionally, subplots were created within each plot to differentiate between the area's interplant and beneath the shrubs of *H. ammodendron*. Considering the atmospheric N deposition rate in the Gurbantunggut Desert [34], two levels of N addition were implemented in the experiment, namely 2.5 gN m$^{-2}$ and 5 gN m$^{-2}$ per year. NH$_4$NO$_3$ was used to administer the N treatment. It was diluted in 15 L of distilled water, which was equivalent to 0.15 mm of rainfall, and evenly sprayed onto the corresponding plots. To facilitate the N absorption in plants, the N application took place after rainfall events, specifically in early April and mid-July, which coincided with the growth phases of ephemerals and annuals, respectively. The control

plots received an equal amount of distilled water. The experimental designs and instrument arrangements were established in 2010, and the background investigation confirmed the absence of heterogeneity among the blocks.

### 2.3. Soil Sampling and Measurements

In late April and early August of 2011, 2014, and 2019, soil samples were obtained from each subplot using soil cores measuring 5 cm in diameter and 5 cm in depth. Each sample consisted of five soil core samples from a depth of 0–5 cm within the subplot. Immediately after collection, the samples were placed in plastic bags and transported on ice to the laboratory, where they were stored at 4 °C for subsequent analysis. To ensure the integrity of microbial activity and avoid any interference from rainfall [35–37], soil sampling took place at least five days after the most recent rainfall event. Within 24 h of sampling, the subsamples from each raw soil sample were utilized to determine soil microbial C utilization, as well as microbial biomass C and N. The remaining samples were employed for phospholipid fatty acid (PLFA) profiles and measurements of soil physicochemical properties.

### 2.4. Soil Properties Measurements

Soil nitrate-N ($NO_3^-$-N) and ammonium-N ($NH_4^+$-N) were extracted from 10 g of fresh soil using 50 mL of a 2 M KCl solution. The concentrations of these components were measured using a continuous-flow ion auto-analyzer (AA3, *BRAN-LUEBBE* Ltd., Hamburger, Germany). Dissolved organic carbon was extracted from 12.5 g of soil using 50 mL of 0.5 M $K_2SO_4$ solution and was subsequently analyzed using a TOC analyzer (multi N/C 3100, Jena, Germany). The soil volumetric water content was determined at a depth of 0–5 cm using a portable TDR ($HH_2$-Delta T Device moisture meter, UK). The soil pH was assessed by employing a glass electrode in a 1:5 soil-to-water ratio (*w/v*).

### 2.5. Net N-mineralization In Situ

The buried core incubation method [38] was used to estimate the net N mineralization rate ($R_mN$) and nitrification rate ($R_{NN}$) in April and August, which represented the wet and dry months in a normal year. Three paired samples (with a distance of 20 cm between the two) were selected both in the interplant and beneath shrubs: one sample was used to determine the initial condition, and the other two were used for in situ incubation. Incubation was performed using perforated PVC tubes (15 cm in height and 6 cm in diameter), which were inserted into soils, and a plastic cap covered the top of the tube to avoid nitrate leaching. After 30 days of incubation, soil in the PVC tubes was taken to the lab for ammonium and nitrate analysis. The N mineralization rate was estimated as the difference in the inorganic nitrogen concentration ($NH_4^+$-N and $NO_3^-$-N) between the initial and the final. The net nitrification rate was estimated by the difference between the initial and final $NO_3^-$-N.

### 2.6. Soil C Mineralization

In addition, soil C mineralization was assessed by measuring the $CO_2$ evolution from fresh soil samples, which were incubated in sealed containers at 60% of their water-holding capacity for 4 h at 25 °C. The $CO_2$ efflux from the soil was determined using a Hewlett-Packard 5890 Series II gas chromatograph, which was equipped with an electron capture detector. The rates of $CO_2$ emissions were calculated using the modified equation described by Lang et al. [39]:

$$F = \frac{V \times \Delta C \times \rho \times 273.15}{(273.15 + T) \times W \times t}$$

Here, *F* represents the $CO_2$ emission rate (μg $CO_2$-C $g^{-1}$ $h^{-1}$), *V* is the volume of the headspace of the microcosms (*L*), $\Delta C$ is the change in gas concentration (ppm) from the start to the end of the incubation, $\rho$ is the density of $CO_2$ under standard conditions

(at normal atmospheric pressure), *T* is the incubation temperature (°C), *W* is the soil dry weight (g), and *t* is the time interval of sampling.

### 2.7. Soil Microbial Biomass and Microbial Community Composition Measurements

Soil microbial biomass C ($C_{mic}$) and N ($N_{mic}$) were determined using the chloroform fumigation extraction method [40]. Paired samples of 20 g of fresh soil were either left unfumigated or fumigated with alcohol-free $CHCl_3$ for 24 h. Subsequently, the samples were extracted with 50 mL of a 0.5 M $K_2SO_4$ solution (at a ratio of 1:2.5 *w/v*). The fumigated and non-fumigated extracts were analyzed for their total C and N using a TOC analyzer (multi N/C 3100, Jena, Germany). The respective biomass of $C_{mic}$ and $N_{mic}$ was calculated using the efficiency factors that were specific to each parameter [40,41].

To evaluate the microbial community's composition, phospholipid fatty acid (PLFA) analysis was conducted. The soil samples without roots and gravels were utilized for the fractionation and quantification of microbial PLFAs following the protocol described by White et al. [42]. In summary, 8 g of soil (dry weight) was used to extract PLFAs in a chloroform-methanol-citrate buffer (at a ratio of 1:2:0.8 *v/v/v*). The lipids were then separated into neutral lipids, glycolipids, and phospholipids using a silicic acid column (Sep-Pak Plus Silica; Waters Corp., Milford, MA, USA). The phospholipids were esterified with HCl methanol (Tokyo Kasei Kogyo Co., Ltd., Tokyo, Japan). The resultant fatty acid methyl esters (FAMEs) were identified using a MIDI peak identification system (Microbial ID Inc., Newark, DE, USA). The peak areas were converted to nmol lipid per gram of dry soil using internal standards (19:0 nonadecanoic methylester). The fatty acid nomenclature proposed by Frostegard et al. [43] was followed. The bacterial markers [44–46] included i14:0, 14:1w5c, i15:0, a15:0, i16:0, 16:1w5c, 16:1w9c, i17:0, a17:0, 17:1w8c, 17:1w9c, cy17:0, cy19:0, 18:1w7c, 20:1w9c, as well as normal saturated PLFAs (14:0, 16:0, 17:0, 18:0, 20:0), which were chosen as bacterial markers. Fungal markers [43,47–49] were represented by unsaturated PLFAs, namely 18:1w9c and 18:2w6c. The ratio of the fungal lipids to bacterial lipids was used to indicate the fungal to bacterial PLFAs [45,50].

For the measurement of herbaceous biomass, five quadrats measuring 1 m × 1 m were established in each plot, with three quadrats in interplant spaces and two beneath the shrubs. In May 2011, 2014, and 2019, all living biomasses within each quadrat were harvested in mid-May. The harvested plants were then weighed after drying in an oven at 65 °C for 48 h. The aboveground biomass of the herbaceous vegetation was expressed in grams per square meter (g m$^{-2}$).

### 2.8. Herbaceous Biomass Measurement

Five 1 m × 1 m quadrats were established in each plot, three in interplant spaces and two beneath shrubs, in early May of 2011, 2014, and 2019. All living biomass was harvested in each quadrat in mid-May, and the harvested plants were weighed after drying in an oven at 65 °C for 48 h. Herbaceous aboveground biomass was expressed in grams per square meter (g m $^{-2}$).

### 2.9. Statistical Analysis

To analyze the effects of the treatment year, site, and N on various parameters, including $R_mN$, $R_{NN}$, $R_{CM}$, soil environmental factors, microbial biomass, and PLFA markers, a three-way ANOVA was conducted. Additionally, the main and interactive effects of the sampling month, N, and site on $R_mN$, $R_{NN}$, and microbial biomass were analyzed using a three-way ANOVA, considering the universal significance of the year. Tukey's post hoc test was applied to identify differences among the N treatments. Furthermore, due to the distinct changing patterns observed between the interplant and beneath shrubs for RCM, microbial biomass, and PLFAs, a one-way ANOVA was performed to assess the impact of the N addition on these variables at each site and sampling month.

The relationship between microbial parameters and $R_mN$, $R_{NN}$, $R_{CM}$, as well as soil and vegetation parameters, was examined using linear regression analysis. Multiple regres-

sion analysis was employed to quantify the combined effects of the soil's environmental, microbial, and vegetation parameters on $R_mN$, $R_{NN}$, and $R_{CM}$. The most influential soil variables were selected using the 'forward selection' procedure within the program.

To delve further into the direct and indirect effects of the N addition and microsite on $R_mN$, $R_{NN}$, and $R_{CM}$, a structural equation modeling approach was utilized. This involved investigating the significant associations between predictors and RmN, $R_{NN}$, and $R_{CM}$ using In-N, SM, $C_{mic}$, $N_{mic}$, PLFA markers, F:B PLFAs, and ephemeral biomass as the variables. The "sem" function in the lavaan package [51] was employed for this analysis. Only significant pathways were retained in the final model, while non-significant pathways were eliminated. All statistical analyses were performed using R version 4.1.0 [52].

## 3. Results

### 3.1. N Mineralization Rate and Nitrification, and C Mineralization Rate in Response to N Addition

N addition significantly promoted $R_mN$ and $R_{NN}$ (Table 1), and they were significantly higher in the N addition of 5 gN m$^{-2}$ yr$^{-1}$ than in the control (Figure 1). $R_mN$ and $R_{NN}$ beneath shrubs were 50% higher than the interplant (Table 1, Figure 1). N and site had interactive effects on $R_mN$ and $R_{NN}$, which were higher beneath shrubs than in the interplant (Table 1, Figure 1). The effects of N addition on $R_{CM}$ differed between sites (Table 2, Figure 2). N addition promoted $R_{CM}$ in the interplant, while the N addition of 5 gN m$^{-2}$ yr$^{-1}$ depressed $R_{CM}$ beneath the shrubs (Table 2, Figure 2).

**Table 1.** Effects of month, N addition treatment, site and two-way interactions of N and site on soil net N mineralization rate ($R_mN$), soil net nitrification rate ($R_{NN}$), microbial biomass C ($C_{mic}$), microbial biomass N ($N_{mic}$) and the ratio of $C_{mic}$:$N_{mic}$ (Bold numbers indicate significant at the 0.05 level, $n = 6$, Three-way ANOVA).

| Year | Factors | Samling Time | | N | | Site | | N × Site | |
|---|---|---|---|---|---|---|---|---|---|
| | | F | p | F | p | F | p | F | p |
| 2011 | $R_mN$ | **16.57** | **<0.001** | **26.25** | **<0.001** | **35.18** | **<0.001** | **6.34** | **0.004** |
| | $R_{NN}$ | **4.43** | **0.04** | **5.4** | **0.008** | 2.03 | 0.16 | 0.03 | 0.96 |
| | $C_{mic}$ | 0.99 | 0.32 | 1.07 | 0.35 | 0.04 | 0.84 | **6.35** | **0.004** |
| | $N_{mic}$ | 0.14 | 0.71 | 1.4 | 0.25 | 3.4 | 0.07 | **8.37** | **<0.001** |
| | $C_{mic}$:$N_{mic}$ | 0.87 | 0.35 | 0.06 | 0.94 | 1.69 | 0.2 | 0.53 | 0.59 |
| 2014 | $R_mN$ | **43.72** | **<0.001** | **28.62** | **<0.001** | **58.27** | **<0.001** | **9.44** | **<0.001** |
| | $R_{NN}$ | **25.29** | **<0.001** | **10.02** | **<0.001** | **35.43** | **<0.001** | **5.94** | **0.004** |
| | $C_{mic}$ | **43.45** | **<0.001** | 0.55 | 0.58 | 0.18 | 0.68 | 2.07 | 0.14 |
| | $N_{mic}$ | **50.32** | **<0.001** | **4.04** | **0.03** | **25.64** | **<0.001** | **12.47** | **<0.001** |
| | $C_{mic}$:$N_{mic}$ | 0.001 | 0.98 | 0.38 | 0.68 | **11.00** | **0.002** | **5.49** | **0.008** |
| 2019 | $R_mN$ | 0.78 | 0.38 | **13.05** | **<0.001** | **19.28** | **<0.001** | 0.77 | 0.46 |
| | $R_{NN}$ | 0.93 | 0.34 | **22.73** | **<0.001** | **62.16** | **<0.001** | **4.2** | **0.02** |
| | $C_{mic}$ | **49.97** | **<0.001** | 0.28 | 0.75 | **4.55** | **0.03** | **4.26** | **0.02** |
| | $N_{mic}$ | **12.29** | **<0.001** | 0.51 | 0.6 | 0.4 | 0.52 | **8.33** | **0.001** |
| | $C_{mic}$:$N_{mic}$ | **26.16** | **<0.001** | 0.28 | 0.75 | 0.89 | 0.34 | 2.1 | 0.13 |

### 3.2. Seasonal Dynamics of Soil Inorganic Nitrogen

N addition significantly increased soil $NO_3^-$-N, and this positive effect was consistent across the seasons and at two microsites ($p < 0.001$, Figure S1). N addition had no impact on soil $NH_4^+$-N ($p = 0.47$, Figure S1). Soil $NO_3^-$-N and $NH_4^+$-N differed between the microsites ($p < 0.001$). Seasonally averaged soil $NO_3^-$-N beneath the shrubs was about 1.5 times higher than that in the interplant (Figure S1). The seasonally averaged soil $NH_4^+$-N increased from 3.3 mg kg$^{-1}$ beneath the shrubs to 14.1 mg kg$^{-1}$ in the interplant (Figure S1).

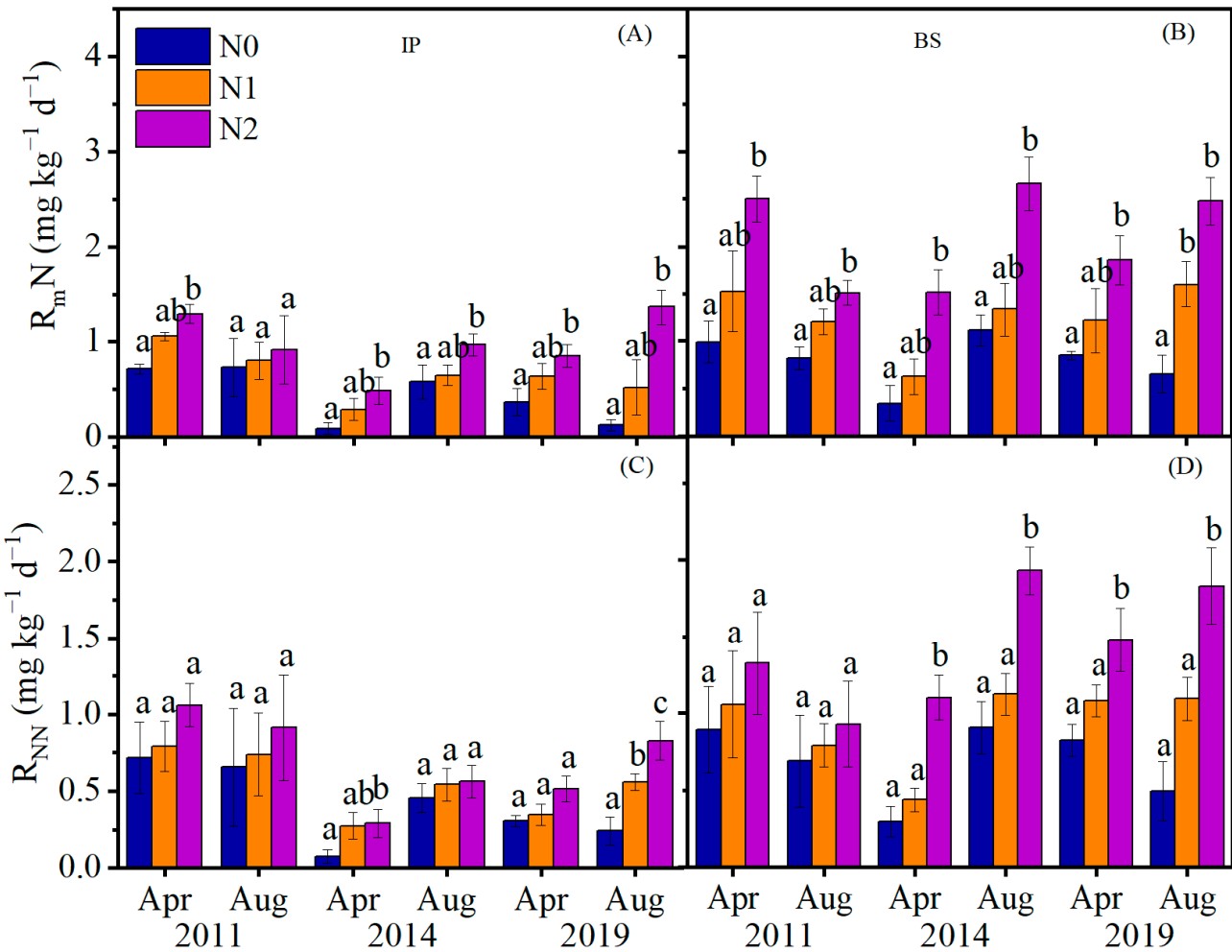

**Figure 1.** N mineralization rate ($R_mN$) at the two microsites of interplant space (IP, (**A**)) and beneath shrub (BS, (**B**)) and soil net nitrification rate ($R_{NN}$) at the two microsites of interplant space (IP, (**C**)) and beneath shrub (BS, (**D**)) under three levels of N treatments (N0, N1 and N2). Different small letters indicate significant differences among the N treatments at $p = 0.05$ level (Tukey's post hoc test). N0: 0 gN m$^{-2}$ yr$^{-1}$ addition; N1: 2.5 gN m$^{-2}$ yr$^{-1}$ addition; N2: 5 gN m$^{-2}$ yr$^{-1}$ addition.

**Table 2.** Effects of year, N addition, site and two-way interactions of N and site on carbon mineralization rate ($R_{CM}$), microbial PLFAs, bacterial PLFAs, fungal PLFAs and the ratio of fungal to bacterial PLFAs (F:B) (Bold numbers indicate significant at the 0.05 level, $n = 6$, Three-way ANOVA).

| Factors | Year | | N | | Site | | N × Site | |
|---|---|---|---|---|---|---|---|---|
| | *F* | *p* | *F* | *p* | *F* | *p* | *F* | *p* |
| $R_{CM}$ | **381.5** | **<0.001** | **100.4** | **<0.001** | **204.72** | **<0.001** | **272.12** | **<0.001** |
| Microbial PLFAs | **143.89** | **<0.001** | 2.99 | 0.06 | **40.6** | **<0.001** | **5.94** | **0.006** |
| Bacterial PLFAs | **409.73** | **<0.001** | **4.83** | **0.01** | **58.36** | **<0.001** | **37.18** | **<0.001** |
| Fungal PLFAs | **335.58** | **<0.001** | 1.55 | 0.23 | 0.3 | 0.58 | **24.24** | **<0.001** |
| F:B | **11.66** | **<0.001** | 0.18 | 0.83 | **60.46** | **<0.001** | **4.68** | **0.01** |

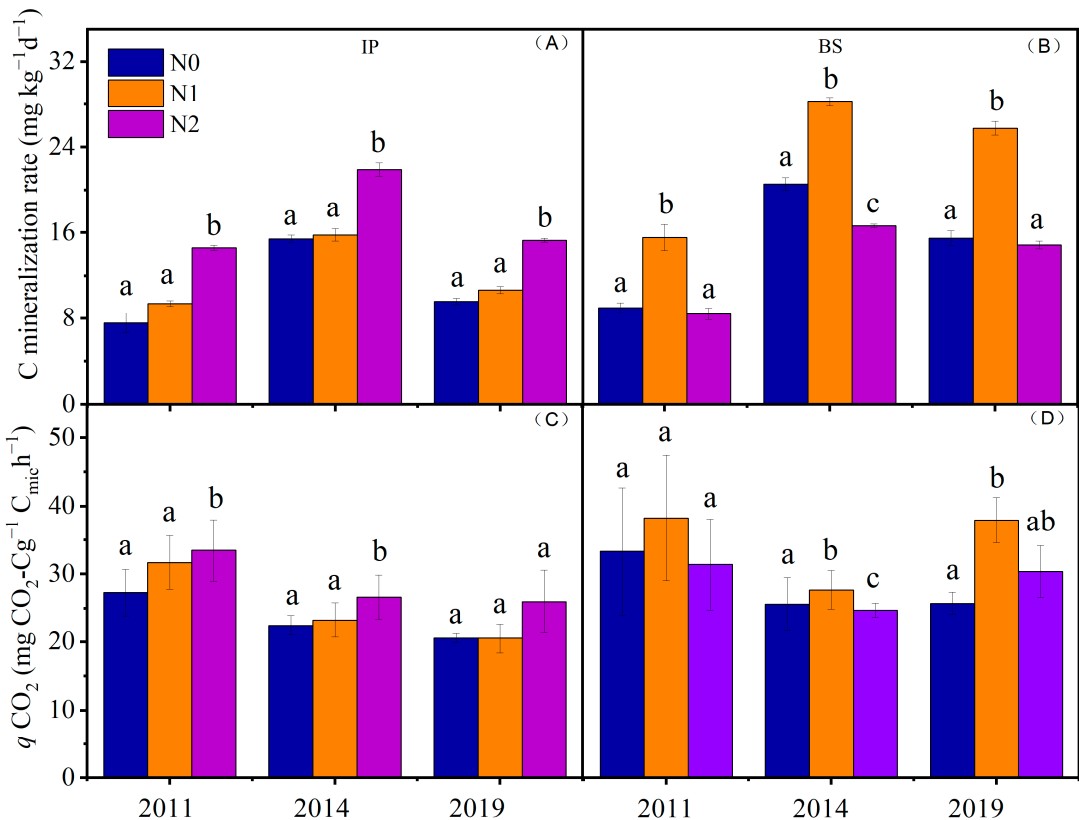

**Figure 2.** Soil organic carbon mineralization at the two microsites of interplant space (IP, (**A**)) and beneath shrub (BS, (**B**)) and microbial quotient at the two microsites of interplant space (IP, (**C**)) and beneath shrub (BS, (**D**)) under three levels of N treatments (N0, N1 and N2). Different small letters indicate significant differences among the N treatments at $p = 0.05$ level (Tukey's post hoc test). N0: 0 gN m$^{-2}$ yr$^{-1}$ addition; N1: 2.5 gN m$^{-2}$ yr$^{-1}$ addition; N2: 5 gN m$^{-2}$ yr$^{-1}$ addition.

### 3.3. Microbial Biomass and Community Composition in Response to N Addition

N addition exerted opposite effects on $C_{mic}$ and $N_{mic}$ between the microsites (Table 2, Figure S2). $C_{mic}$ and $N_{mic}$ in the interplant consistently increased with N addition, from 0.34 g kg$^{-1}$ ($C_{mic}$) to 0.03 g kg$^{-1}$ ($N_{mic}$) and in the control to 0.45 g kg$^{-1}$ ($C_{mic}$) and 0.05 g kg$^{-1}$ ($N_{mic}$) in N addition to 5 gN m$^{-2}$ yr$^{-1}$; whereas $C_{mic}$ decreased from 0.5 mg g$^{-1}$ to 0.39 mg g$^{-1}$, $N_{mic}$ decreased from 0.06 mg g$^{-1}$ to 0.04 mg g$^{-1}$. As a result, the ratio of $C_{mic}$:$N_{mic}$ increased beneath the shrubs and decreased in the interplant (Table 2, Figure S2).

Consistently, the total, bacterial, and fungal PLFAs between the two microsites showed opposite responses to the N addition; they increased beneath the shrubs and decreased in the interplant under N addition (Table 2, Figure S3). In general, the total, bacterial, and fungal PLFAs in the N addition of 2.5 gN m$^{-2}$ yr$^{-1}$ were higher than the control. However, they were lower in the N addition of 5 gN m$^{-2}$ yr$^{-1}$ (Figure S3). The responses of F:B PLFAs to N addition depended on the site (Figure S3), with an increase in the interplant (F = 2.39, $p = 0.09$) and no change beneath the shrubs (F = 2.22, $p = 0.11$).

### 3.4. Factors Determining Soil N and C Mineralization

$R_mN$ was correlated with $N_{mic}$ and the ratio of $C_{mic}$:$N_{mic}$ in the interplant, whereas it showed no correlation with the microbial biomass beneath shrubs (Figure 3). $R_{NN}$ was significantly related to microbial biomass and MBC:$N_{mic}$ in the interplant (Figure 3). Consistent with $R_mN$, $R_{NN}$ was independent of the microbial biomass beneath shrubs. $R_{CM}$ was significantly related to $C_{mic}$ and $N_{mic}$ (Figure 3), including the total and fungal PLFAs in interplant (Figure 4), whereas it was significantly related to MBC, the total PLFAs, and F:B PLFAs beneath the shrubs (Figure 4).

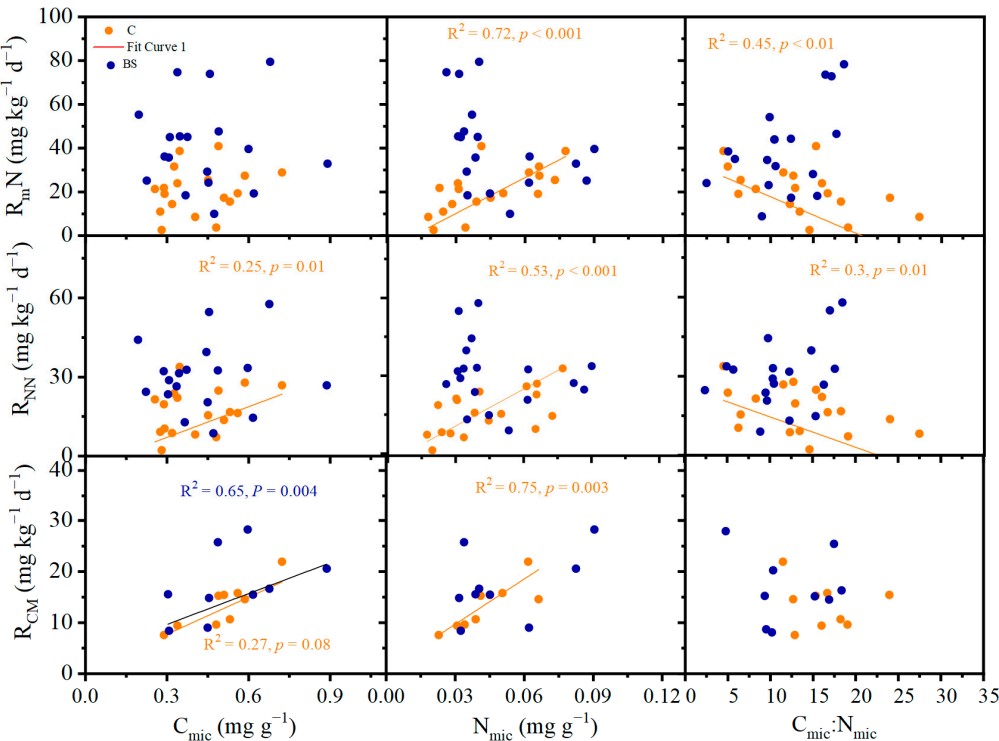

**Figure 3.** Correlations of microbial biomass C ($C_{mic}$), N ($N_{mic}$) and the ratio of microbial biomass C to N ($C_{mic}$:$N_{mic}$) with soil net N mineralization rate ($R_mN$), nitrification rate ($R_{NN}$) and C mineralization ($R_{CM}$) in the interplant space (IP) and beneath shrub (BS). Correlation coefficients and associated *p* values were based on Pearson correlation tests.

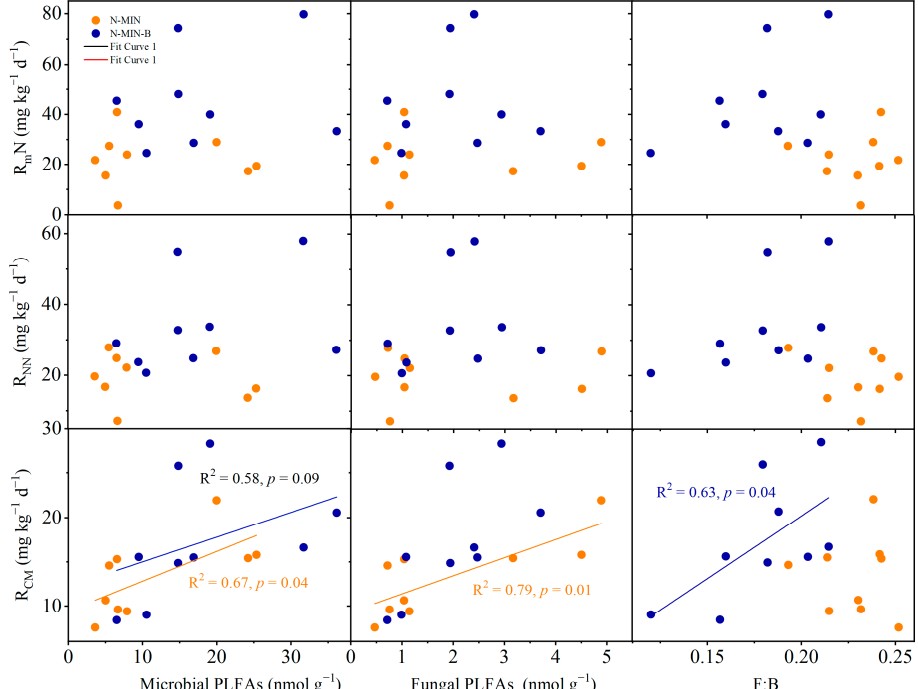

**Figure 4.** Correlations of microbial PLFAs, fungal PLFAs, and the ratio of fungal to bacterial PLFAs (F:B) with a soil net N mineralization rate ($R_mN$), nitrification rate ($R_{NN}$) and C mineralization ($R_{CM}$) in the interplant space (IP) and beneath shrub (BS). Correlation coefficients and associated *p* values were based on Pearson correlation tests.

$C_{mic}$ was correlated with In-N in the interplant space, soil volumetric water content (SVWC) at the two microsites, ephemeral biomass in interplant space, and soil dissolved carbon (DC) beneath shrubs, microbial PLFAs was correlated with In-N in the interplant space, SVWC and ephemeral biomass beneath shrubs, and F:B PLFAs was correlated with In-N in the interplant space (Figure S4).

Structural equation modeling demonstrated that N addition could directly change $R_mN$ and $R_{NN}$ by altering the soil moisture, microbial biomass, microbial structure (F:B), and the ratio of $C_{mic}:N_{mic}$ (Figure 5). Among these controlling factors, soil moisture exerted the highest impact on both $R_mN$ and $R_{NN}$. N addition directly changed $R_{CM}$ (Figure 5). Indirectly, it changed DOC and $N_{mic}$ to impact $R_{CM}$ (Figure 5). DOC exerted more significant impacts on $R_{CM}$ than $N_{mic}$ and In-N (Figure 5).

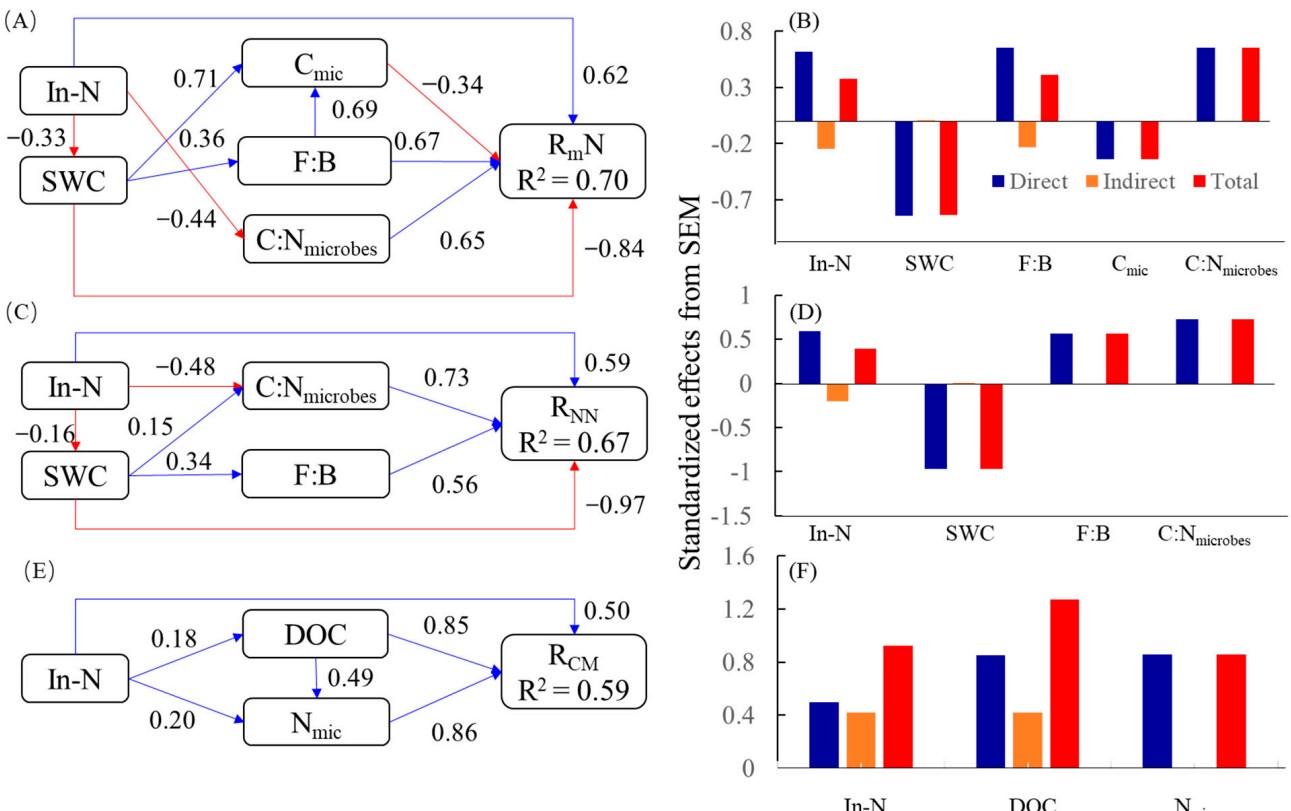

**Figure 5.** Structural equation modeling of the direct and indirect effects of soil and microbial properties on the soil net N mineralization rate ($R_mN$, (**A**,**B**)), soil net nitrification rate ($R_{NN}$, (**C**,**D**)), and soil C mineralization rate ($R_{CM}$, (**E**,**F**)). The total effect (TE) is the sum of direct (DE) and indirect (IE) effects. $R^2$ indicates the proportion of variation in the independent variable explained by the regression model in each case. IE was evaluated as significant or not using a simple linear regression between both variables associated in each case. *In-N* soil inorganic nitrogen, *DOC* dissolved organic carbon; *SWC* soil water content, $C_{mic}$ microbial biomass C, $N_{mic}$ microbial biomass N, $(C:N)_{micorbes}$ ratio of $C_{mic}:N_{mic}$, F:B the ratio of fungi to bacterial PLFAs.

## 4. Discussion

### 4.1. Effects of N Addition on Soil Net N Mineralization and Nitrification Rate

Soil nitrogen mineralization and nitrification rates changed markedly under N addition both in the interplant and beneath shrubs in the temperate desert. Although the applied N in this study was relatively lower compared with other studies [5,6,28,53], N addition still promoted $R_mN$ and $R_{NN}$. This was consistent with some empirical studies [16,54–58]. Soil inorganic nitrogen promoted $R_mN$ and $R_{NN}$, and the increased soil microbial biomass in the interplant significantly increased microbial decomposition, which

could be evidenced by the positive correlations between microbial biomass and $R_mN$ and $R_{NN}$. However, because of the depression of N addition on microbial growth, microbial biomass did not exert significant influences on $R_mN$ and $R_{NN}$ beneath the shrubs. The disparity of microbial responses between microsites could be explained by the opposite patterns of DOC in response to N addition. In addition, although the average soil moisture was lower beneath the shrubs, the soil experienced frequent wet-dry cycles due to stemflow and the evapotranspiration of shrubs [59]. Frequent wet-dry cycles could promote the release of N embedded in soil particles and aggregates, increasing the N transformation from an organic to an inorganic state [59,60]. Thus, $R_mN$ and $R_{NN}$ were stimulated by N addition, though microbial activities were decreased beneath the shrubs.

The average soil C:N ratio was 34, which was significantly higher than the $C_{mic}:N_{mic}$ ratio ($14 \pm 4$), resulting in a lower microbial decomposition. N addition could make soil organic matter more decomposable by decreasing the soil C:N ratio, which, in turn, decreased microbial N accumulation and increased $R_mN$ and $R_{NN}$. Thus, the persistent increase in soil inorganic nitrogen led to the highest $R_mN$ and $R_{NN}$ in the last investigation. However, a persistent N addition could decrease the soil pH and depress microbial activities [61–63], which could subsequently decrease $R_mN$ and $R_{NN}$. Although the pH was 10.43 beneath the shrubs and 10.08 in the interplant at the beginning of N treatment, N addition did not lead to soil acidification.

### 4.2. Effects of N Addition on Soil C Mineralization

The N addition of 5 gN m$^{-2}$ yr$^{-1}$ depressed $R_{CM}$ beneath the shrubs, suggesting that N was not the sole factor limiting the soil organic matter decomposition. This result, together with the decreased microbial biomass and PLFAs beneath the shrubs, implied that the N addition could decrease $R_{CM}$ by inhibiting microbial activities. More importantly, increased $C_{mic}:N_{mic}$ and F:B suggested that N addition had a preference for promoting fungal growth. Given that the fungal hyphae can absorb soil nitrogen, fungi can better avoid N accumulation relative to bacteria [63,64], and the variation in the microbial community composition might be responsible for the different changes of $R_{CM}$ with $R_mN$ and $R_{NN}$.

A novel perspective of this study is that N addition can exert opposite effects on $R_{CM}$ and $R_mN$ beneath the shrubs while exerting consistent effects on them in the interplant. The inconsistent responses of $R_{CM}$ to the N addition between the two sites indicate the importance of N availability to microbial respiration, which is not equal between the interplant and beneath shrubs. Considering the significant correlations of DOC with $C_{mic}$ and the total PLFAs, the depression of $R_{CM}$ under high N addition (N2) indicated that C might be an alternative limiting factor for microbial activity beneath the shrubs; on the other hand, increased $R_{CM}$ under low N addition (N1) suggested the priming effect of N on $R_{CM}$ [65–67]. In the pooled data of the two microsites as a whole, $R_{CM}$ was controlled by soil nutrients and microbial biomass. Although microbial biomass beneath the shrubs was higher than the interplant, microbial biomass and PLFAs were reduced by N addition beneath the shrubs, which was partly responsible for the decreased $R_{CM}$. The positive correlation between F:B and $R_{CM}$ also indicated that variations in the microbial activity under N addition resulted from increasing microbial biomass and a shifting microbial community structure.

### 4.3. The Opposite Responses of Microbial Characteristics between Interplant and Beneath Shrubs

In contrast to our hypothesis, N and the microsite (fertile islands) had no interactive effects on $R_mN$ and $R_{NN}$, whereas they interacted to affect microbial biomass and PLFAs. N addition promoted $C_{mic}$ and $N_{mic}$, the total, fungal, and bacterial PLFAs in the interplant, suggesting that soil microbial communities were N-limited in the interplant. However, the negative responses of microbial biomass and PLFAs to N addition suggested that soil N might be saturated with microbes beneath the shrubs. This perspective could be evidenced by N addition studies in N-saturation ecosystems [6,61,68], showing that there was a certain threshold value for the N content for microbial communities [5–7,13,69]. Soil microbial

activity increases non-linearly with a gradient of N addition [6,7,70–72], and this threshold varies in different ecosystems [5,7,70]. In addition, C limitation may be responsible for decreased C mineralization beneath the shrubs under N addition.

$R_{CM}$ was related to the ratio of F:B, suggesting that the microbial community structure can influence $R_{CM}$. The reduced soil N availability may enhance the nutrient competition between plants and soil microbes, resulting in a nutrient limitation to microbes beneath the shrubs [16,73,74]. Under low nutrients, fungi relocate nutrients through the hypha and recycle nutrients (especially inorganic N) via cytoplasm translocation. This enhances their competitive ability over the bacteria to exploit available nutrients [75]. Thus, decreased soil N availability likely contributed to the enhanced fungi dominance in microbial communities beneath the shrubs.

### 4.4. Implications for N Cycling in Deserts

This study has two implications for N dynamics in deserts. First, N addition can increase soil inorganic N, suggesting an accumulation of N in the soil profiles under N deposition. Increased N availability, accompanied by relatively higher available soil C beneath the shrubs, may increase the soil organic matter decomposition and inorganic N content; this can subsequently promote shrub growth and alter plant community composition. Second, the soil showed high alkalinity and salinity in this desert, where nitrification could exceed denitrification [76,77], and a high $R_mN$ and $R_{NN}$ under N addition could be accompanied by increased N loss via $N_2O$ emissions. Especially physical processes such as $NH_3$ volatilization also contribute to soil N loss under drought [78,79]. In addition, interplant soils showed a substantially lower $R_mN$ and $R_{NN}$ and higher $R_{CM}$ compared to the soil beneath shrubs. These differences across a landscape scale highlight the concept of 'fertile islands' when governing the spatial distribution of desert resources and microbial communities [27,79,80].

The C limitation to microbial N cycling may occur as the soil N content arrives at a high level. Given that the soil beneath shrubs takes the main proportion of desert C stock, decreased $R_{CM}$ beneath shrubs implies that the soil C stock increases under high N deposition. Therefore, soils beneath old shrubs and dead-standing plants can be more sensitive to N decomposition in deserts.

### 5. Conclusions

The addition of nitrogen can enhance soil nitrogen mineralization and nitrification, but the impact on soil carbon mineralization is influenced by the amount of nitrogen added, the available carbon content in the soil, and moisture levels. The different responses of microbial communities to nitrogen addition in different microsites indicate that the effects of "fertile islands" created by the nutrients and microbial biomass are important when estimating the feedback of carbon and nitrogen cycling in desert ecosystems in response to projected nitrogen deposition.

The rate of nitrogen mineralization consistently increased with nitrogen addition, while the carbon mineralization rate was only slightly promoted in the interplant areas. This suggests that the decoupling of carbon and nitrogen cycling may occur in deserts with relatively higher rates of nitrogen deposition. Microbial biomass carbon, total phospholipid fatty acids (PLFAs), and the ratio of fungal to bacterial PLFAs were suppressed by the nitrogen addition under shrubs, and these were closely related to the carbon mineralization rate, indicating the critical roles of the microbial biomass and community structure in regulating carbon cycling. A small addition of nitrogen increased soil nitrogen mineralization and nitrification, but its effects on soil carbon mineralization depended on the amount of nitrogen added, the dissolved organic carbon content in the soil, and moisture levels.

**Supplementary Materials:** The following supporting information can be downloaded at: https://www.mdpi.com/article/10.3390/f14061154/s1, Figure S1: Soil nitrate ($NO_3^-$-N) and ammoniacal ($NH_4^+$-N) content in two microsites of interplant space and beneath the shrub under three levels of N treatments (N0, N1 and N2); Figure S2: Soil microbial biomass carbon ($C_{mic}$), nitrogen ($N_{mic}$), the

ratio of soil microbial biomass carbon to nitrogen ($C_{mic}:N_{mic}$) in two microsites of interplant space and beneath shrub under three levels of N treatment (N0, N1 and N2); Figure S3. Soil microbial phospholipid fatty acids (PLFAs), bacterial PLFAs, fungal PLFAs and soil fungal to bacterial PLFAs (F:B) in two microsites of interplant space and beneath the shrub under three levels of N treatments (N0, N1 and N2); Figure S4: Correlations of soil inorganic nitrogen (In-N), soil volumetric water content (SVWC), ephemeral biomass and soil dissolved carbon (DC) with microbial biomass C ($C_{mic}$), microbial PLFAs and the ratio of fungal to bacterial PLFAs (F:B); Table S1: Soil and vegetation properties in the interplant (IP) and beneath shrubs (BS) of the study site.

**Author Contributions:** Conceptualization, Y.C.; methodology, Y.C.; formal analysis, H.L. and Y.C.; investigation, H.L., H.S., Y.G. and Y.C.; writing—original draft preparation, Y.C.; writing—review and editing, Y.C.; visualization, Y.G.; project administration, H.L. and Y.C; funding acquisition, Y.C. All authors have read and agreed to the published version of the manuscript.

**Funding:** This research was funded by the Chinese National Natural Scientific Foundation, grant numbers 40971156 and 32070477.

**Data Availability Statement:** Data are available on request due to restrictions.

**Conflicts of Interest:** The authors declare no conflict of interest.

## Abbreviations

| | |
|---|---|
| $C_{mic}$ | Microbial biomass carbon |
| $N_{mic}$ | Microbial biomass nitrogen |
| $C_{mic}:N_{mic}$ | The ratio of microbial biomass carbon to nitrogen |
| $R_mN$ | Nitrogen mineralization rate |
| $R_{NN}$ | Nitrogen nitrification rate |
| $R_{CM}$ | Carbon mineralization rate |
| In-N | Inorganic nitrogen |
| SM | Soil moisture |
| F:B PLFAs | The ratio of fungal to bacterial PLFAs |

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
