# Peer review of "Microsite Determines the Soil Nitrogen and Carbon Mineralization in Response to Nitrogen Addition in a Temperate Desert"

_forests, doi:10.3390/f14061154_

Round 1

Reviewer 1 Report

                     1.         Line 20. Change to “various responses”.

2.         Line 39. I wonder if a term such as “temperate deserts” exists. In general, temperate climates are defined as environments with moderate rainfall spread across the year or portion of the year with sporadic drought, mild to warm summers and cool to cold winters. Accordingly, the word “temperate” does not belong here.

3.         Line 42. Change “discrepancies” to “differences”.

4.         Lines 52-54. Could you clarify a little better what you mean?

5.         Line 74. As mentioned above the word “temperate” does not belong here.

6.         Lines 83-86. You have to make a separate paragraph under the title “Soil analysis” and insert this piece of information there.

7.         Line 90. Explain the acronyms “CK, N1 and N2”.

8.         Line 103. Table 1 belongs to the Results Section.

9.         Line 136. Why do you consider nitrification more important than ammonification?

10.     Figures 3 and 4. Figures 3 and 4 are incomprehensible. What are IP and BS? Another question that arises is that we have only one R2. From  both figures it seems that we have two correlations, one with the black dots and the other with the orange dots. However, we have only one R2.

11.     In the Discussion and Conclusion sections make it clearer which comparison concerns the interplant areas and which the near the shrubs ones.

The quality of the English language is adequate

Author Response

  1. Line 20. Change to “various responses”.

Response: we have revised “” to “various responses”

  1. Line 39. I wonder if a term such as “temperate deserts” exists. In general, temperate climates are defined as environments with moderate rainfall spread across the year or portion of the year with sporadic drought, mild to warm summers and cool to cold winters. Accordingly, the word “temperate” does not belong here.

Response:thank you for the comment, the “temperate deserts” have been widely used in studies, and we have listed some of these studies below:

Yang F , Zhou G . Characteristics and modeling of evapotranspiration over a temperate desert steppe in Inner Mongolia, China[J]. Journal of Hydrology, 2011, 396(1-2):139-147.

Fang, Li, Wenzhi, et al. The Response of Aboveground Net Primary Productivity of Desert Vegetation to Rainfall Pulse in the Temperate Desert Region of Northwest China[J]. PLoS ONE, 2013.

Fang B , Zhou G , Wang F , et al. Partitioning soil respiration in a temperate desert steppe in Inner Mongolia using exponential regression method[J]. Soil Biology & Biochemistry, 2010, 42(12):2339-2341.

Zhang, Liu, JL, et al. Spatial and environmental determinants of plant species diversity in a temperate desert[J]. J PLANT ECOL, 2016, 2016,9(2)(-):124-131.

  1. Line 42. Change “discrepancies” to “differences”.

Response: we have revised “discrepancies” to “differences”.

  1. Lines 52-54. Could you clarify a little better what you mean?

 Response: we have rewritten this sentence.

  1. Line 74. As mentioned above the word “temperate” does not belong here.

Response: the climate in the study site is temperate

  1. Lines 83-86. You have to make a separate paragraph under the title “Soil analysis” and insert this piece of information there.

Response: Thank you for the comment. The vegetation and soil properties were all from previous studies, so we cited papers here.

  1. Line 90. Explain the acronyms “CK, N1 and N2”.

 Response: we have explained the acronyms “CK, N1 and N2” here.

  1. Line 103. Table 1 belongs to the Results Section.

Response: thank you for the comment, we have revised here.

  1. Line 136. Why do you consider nitrification more important than ammonification?

Response: we have revised here.

  1. Figures 3 and 4. Figures 3 and 4 are incomprehensible. What are IP and BS? Another question that arises is that we have only one R2. From both figures it seems that we have two correlations, one with the black dots and the other with the orange dots. However, we have only one R2.

Response: Thank you for your comment, we only fitted the significant ones.

  1. In the Discussion and Conclusion sections make it clearer which comparison concerns the interplant areas and which the near the shrubs ones.

Response: we have checked and made revision throughout the discussion and conclusion sections.

Reviewer 2 Report

Dear authors.

The article is good, but editing and adding text to the summary, Conclusions and tables are needed.

Summary and conclusions need to be expanded

  It is necessary to add the following:

Two levels of nitrogen application were used. It is necessary to draw a conclusion, which one is more effective?

  What are the conclusions with different duration of aftereffect of nitrogen fertilizers for 9 years?

What fertilizers were applied?

On top of the soil is a layer of sand. Which layer was used for analyzes - sand  or soil proper?

Solonetz is alkaline soils with exchangeable sodium and need to be reclaimed

There are many abbreviations and they are very difficult to understand. It's better to write6

 Ð¡mic Nmic RmN Rms RNN

Explain why fertilizers act differently under bushes and between bushes?

Strengthening 1.Nitrogen mineralization and 2.soil nitrification are two different processes. One is negative. and the second one is positive. This must be reflected in the article and the mechanism explained.

Table 1. What are F and P?

Table 2. shorten the table title. It contains F twice.

Drawing . 1 Write what A, B, C and D stand for or remove them

Rice. 2. shorten the name. What is common in the title in Fig. 1 and fig. 2, remove from fig. 2 and write Legend: - see fig. 1. and also with the letters A.V.S.D, as in Fig. 1

For all figures and tables, the title must be abbreviated.

Table .1 of -supplementary. WRite for what depth all properties are defined, because there was sand on top of the soil

It is necessary to write how in demand and relevant these studies of the authors are. i.e. how many such soils are there in China?

Good luck

15/05/2023

Author Response

Dear authors.

The article is good, but editing and adding text to the summary, Conclusions and tables are needed.

Summary and conclusions need to be expanded

It is necessary to add the following:

Two levels of nitrogen application were used. It is necessary to draw a conclusion, which one is more effective?

Response: we have elucidated the effects of N addition amount on C and N mineralization in the conclusion section.

What are the conclusions with different duration of after effect of nitrogen fertilizers for 9 years?

Response: we have elucidated the consequences of N addition on C and N mineralization after 9 years of treatment in the conclusion section.

What fertilizers were applied?

Response: we have illustrated the applied N in the material and method section.

On top of the soil is a layer of sand. Which layer was used for analyzes - sand or soil proper?

Response:We sampled the top 5 cm for soil property analysis. We have revised in the material and method section, as well as Table S1.

Solonetz is alkaline soils with exchangeable sodium and need to be reclaimed

Response:we have reclaimed this in the site description section.

There are many abbreviations and they are very difficult to understand. It's better to write Сmic Nmic RmN Rms RNN

Response:We have revised based on the comment throughout our paper.

Explain why fertilizers act differently under bushes and between bushes?

Response:Thank you for the comment. Microbial biomass and the ratio of fungi to bacterial PLFAs responded differently to N addition between two microsites. We have linked the changed C mineralization with microbial biomass and community structure indicated by F:B in the discussion section.   

Strengthening 1.Nitrogen mineralization and 2.soil nitrification are two different processes. One is negative. and the second one is positive. This must be reflected in the article and the mechanism explained.

Response:we have mentioned this in the discussion section.

Table 1. What are F and P?

Response:we have revised and illustrated in the table title.

Table 2. shorten the table title. It contains F twice.

Response:we have checked and revised the table title.  

Drawing . 1 Write what A, B, C and D stand for or remove them

Response:We have illustrated the meaning of A, B, C and D in figure title.

Rice. 2. shorten the name. What is common in the title in Fig. 1 and fig. 2, remove from fig. 2 and write Legend: - see fig. 1. and also with the letters A.V.S.D, as in Fig. 1

Response:We have illustrated the meaning of A, B, C and D in figure title.

For all figures and tables, the title must be abbreviated.

Response:We have revised to be abbreviation

Table .1 of -supplementary. Write for what depth all properties are defined, because there was sand on top of the soil

Response:Thank you for the comment, the measurements for soil properties were conducted at 0-5 cm soil depth and we have also illustrated it in the material and method section.

It is necessary to write how in demand and relevant these studies of the authors are. i.e. how many such soils are there in China?

Response:We have illustrated the necessary of the study in the introduction. In the introduction section, we stated that “The global area of deserts is about 2.77 × 109 ha, which accounts for about 20 % of the worldwide land area (Beer et al., 2010). Deserts store 208Pg organic carbon to a depth of 3 m, accounting to 8.9% of the C stock in global soil organic C stock (Jobbàgy &Jackson, 2000).”
